# Feature Shift Detection: Localizing Which Features Have Shifted via Conditional Distribution Tests

**Sean M. Kulinski**           **Saurabh Bagchi**           **David I. Inouye**

School of Electrical and Computer Engineering
Purdue University
{skulinsk,sbagchi,dinouye}@purdue.edu

## Abstract

While previous distribution shift detection approaches can identify if a shift has occurred, these approaches cannot localize which specific features have caused a distribution shift—a critical step in diagnosing or fixing any underlying issue. For example, in military sensor networks, users will want to detect when one or more of the sensors has been compromised, and critically, they will want to know which specific sensors might be compromised. Thus, we first define a formalization of this problem as multiple conditional distribution hypothesis tests and propose both non-parametric and parametric statistical tests. For both efficiency and flexibility, we then propose to use a test statistic based on the density model score function (*i.e.*, gradient with respect to the input)—which can easily compute test statistics for all dimensions in a single forward and backward pass. Any density model could be used for computing the necessary statistics including deep density models such as normalizing flows or autoregressive models. We additionally develop methods for identifying when and where a shift occurs in multivariate time-series data and show results for multiple scenarios using realistic attack models on both simulated and real world data. [1]

## 1   Introduction and Motivation

Adversarial attacks on sensor streams that feed into complex machine learning systems can create serious vulnerabilities. To support decision making in an adversarial setting, it is critical to identify *when* an adversarial attack has started and in a multi-sensor environment, *which* sensors have been compromised. The identification of which particular sensors have been compromised, a process referred to as *"localization"*, is critically important for correction either by physical intervention or hardware reset. While there has been much work on anomaly detection and some on distribution shift [2, 19, 27], the task of identifying or explaining the anomalous behavior or distribution shift remains a relatively open problem. We consider primarily a sensor network attack, where the adversary is able to observe and to control one or more nodes, which are then denoted as "compromised (or attacked) sensors". While most previous adversarial or anomaly detection methods focus on identifying an attack from a single sample [9, 22, 31], we focus on detecting sequences of samples that show anomalous behavior—which we argue is more natural for sensor streams and enables detection of powerful adversaries. For example, consider Figure 1a, which shows that taken in isolation a single sensor reading may not appear as anomalous, but a sequence of readings may appear to deviate from a notional correct distribution, thus identifying the attack. ***Thus, we frame our core problem as distribution shift detection via hypothesis testing instead of standard anomaly detection.*** We make the significant observation that dependencies between samples within a sensor (*i.e.*, captured by their marginal distribution) and dependencies across sensors (*i.e.*, captured by conditional distributions of

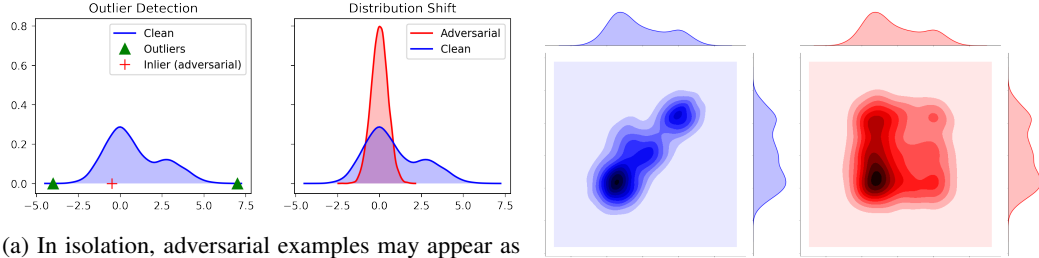

(a) In isolation, adversarial examples may appear as inliers rather than outliers (left), but when when modeling multiple adversarial examples, a shift from the original distribution can be detected because of the relationships between adversarial examples (right)—*i.e.*, adversarial examples are coming from a different distribution. While this illustration is only 1D, the same intuition can be applied to high-dimensional data.

(b) If an adversary compromises one sensor, they may try to match the marginal distribution of that sensor (*e.g.*, by looping the sensor data from a previous day). However, a shift in distribution can be detected by joint analysis across sensors because the adversary does not know the dependencies between all sensors.

Figure 1: (Left subfigure) Rationale for detecting a single compromised sensor using distribution shift detection rather than individual sensor readings. (Right subfigure) Rationale for using conditional distributions of sensor values for correlated sensors, rather than only marginal distributions, to detect attacks against single or multiple sensors.

one sensor, conditioned on other sensor streams) enable the detection of powerful adversarial attacks that try to avoid detection. For example, consider Figure 1b, where the adversary can match the *marginal distribution* of sensor values for a single compromised sensor, but she may not be able to match the *conditional distribution* of the sensor values, conditioned on that of other sensors that have *not* been compromised. This raises the bar on the attack in that *all* sensors which are conditioned on one another will have to be compromised to defeat the detection.

One of the key challenges of our approach is computational efficiency (especially in an online learning setting) because we have to run a statistical test at every time point and for every feature in the multi-sensor setting (each sensor contributes one feature). To address this challenge, we develop a novel test statistic based on Fisher divergence that could be computed in an online manner for all features *simultaneously*. This test statistic is easy to calculate even for complex deep neural density models. Further, it has the desirable properties that it allows us to consider all features simultaneously and thus scales well with the number of sensors—roughly reducing the amount of computation by $O(d)$ where $d$ is the number of sensors since we can compute the needed test statistic for all features simultaneously in one forward and backward pass of a density model. We extend our approach to handle time-series data where a straightforward application of our earlier approach enables us to identify when a sensor has been compromised and in a sensor network attack case, which set of sensors is compromised. We summarize our contributions as follows:

1. We motivate and define the problem of feature shift detection for localizing which specific sensor values (or more broadly which features) have been manipulated.
2. We define conditional distribution hypothesis tests and use this formalization as the key theoretical tool to approach this problem. We propose both model-free and model-based approaches for this conditional test and develop practical algorithms.
3. In particular, we propose a score-based test statistic (*i.e.*, gradient with respect to input) inspired by Fisher divergence but adapted for a novel context as a distribution divergence measure. This test statistic scales to higher dimensions and can be applied to modern deep density models such as normalizing flows and autoregressive models.
4. We propose simple extensions to time series data so that we can identify both when and where a security attack has occurred.

**Notation.** Let $p$ denote a reference distribution and $q$ be a query or test distribution, which may be the same or different than $p$. Let $X$ denote samples from $p$ and $Y$ denote samples from $q$ similarly. For model-based methods, we will denote the models as $\hat{p}$ and $\hat{q}$ for distributions $p$ and $q$ respectively. We will denote vectors by boldcase letters (e.g. $\boldsymbol{x}$) and single elements by non-bold face letters (*e.g.*, $x_1, x_3, x_j$). We will denote $\boldsymbol{x}_{-j}$ as the feature vector that includes all features except for the $j$-th feature (*e.g.*, $\boldsymbol{x}_{-2} = [x_1, x_3, x_4, \ldots, x_d]$, where $d$ is the number of features. The set of all possible $\boldsymbol{x}$, $x_j$, and $\boldsymbol{x}_{-j}$ will be denoted by $\mathcal{X}$, $\mathcal{X}_j$ and $\mathcal{X}_{-j}$ respectively. We will denote a set of samples (formed as a matrix) as a capital letter, *i.e.*, $X = [\boldsymbol{x}_1, \boldsymbol{x}_2, \ldots, \boldsymbol{x}_m]^T \in \mathbb{R}^{n \times d}$, where $n$ is the number

of samples and $d$ is the number of dimensions (which is equal to the number of sensors in our setup). The set of compromised sensors will be denoted by $\mathcal{A} \subset \{1, 2, \cdots, d\}$, and its complement will be denoted by $\overline{\mathcal{A}}$.

## 1.1 Related Works

Distribution shift have been considered in many contexts (see [6, 19] for survey articles). Given a reference distribution, $p$, and a query distribution, $q$, distribution shift is the problem of detecting when $q$ is different from $p$. While there exist standard methods for univariate detection, the multivariate case is still maturing [25]. Generally, there are different types of distribution shift including (1) covariate shift in which $q(\boldsymbol{x}) \neq p(\boldsymbol{x})$ but the conditional of the target given the covariates is the same, *i.e.*, $q(y|\boldsymbol{x}) = p(y|\boldsymbol{x})$, (2) label shift in which $q(y) \neq p(y)$ but $q(\boldsymbol{x}|y) = p(\boldsymbol{x}|y)$, and (3) concept drift in which either (a) $q(\boldsymbol{x}) = p(\boldsymbol{x})$ but $q(y|\boldsymbol{x}) \neq p(y|\boldsymbol{x})$ or (b) $q(y) = p(y)$ but $q(\boldsymbol{x}|y) \neq p(\boldsymbol{x}|y)$ [16, 18, 20]. For classification settings, a hierarchical hypothesis testing framework was developed for detecting concept drifts [29]. More recently, joint distribution shift detection via two-sample hypothesis testing for high-dimensional data has been explored in [25] motivated by detecting shifts for ML pipelines. It empirically compares many different methods including multivariate hypothesis tests (via Maximum-Mean Discrepancy [8]), a combination of multiple univariate hypothesis tests (via multiple marginal KS statistic tests combined via the Bonferonni correction [3]), and possible dimensionality reduction techniques prior to hypothesis testing. The key difference between our work and previous work (including [25]) is that we are concerned with *localizing* which feature or sensor (or set of sensors) is responsible for the distribution shift rather than merely determining if there is a shift. Thus, most previous methods are inapplicable to our problem setting because they either reduce the dimensionality (thereby removing information needed for localization) or perform a joint distribution test (via MMD or similar), which cannot localize which features are compromised. Additionally, most prior work considers the supervised setting whereas we are more interested in the unsupervised setting for detecting compromised sensors.

We will leverage insights from score-based methods [11] in our proposed test statistic for scalability. Score matching was originally proposed as a novel way to estimate non-normalized statistical models because the score is independent of the normalizing constant [11, 30]. This method of estimation can be seen as minimizing the Fisher Divergence [26], which is a divergence metric based on the score function [17]. Score matching has also been generalized to the discrete case [12, 21].

## 2 Feature Shift Detection

In our context, we assume that various sensors are *controllable* (*i.e.*, the values can be manipulated) by an adversary. Our key assumption is that an adversary can only control a subset of sensors in the network—otherwise, an adversary could manipulate the entire system and there would be no hope of detecting anything from sensor data alone. We also assume that there is no causal relationship between sensor values. Our goal is to identify which subset of sensors have been compromised or attacked. Formally, our problem statement can be defined as:

**Definition 1. [Feature Shift Detection Problem Formulation]** *We are given two sets of samples from $p$ and $q$, denoted $X$ and $Y$ respectively. Now, identify which subset of features (if any) have been attacked, denoted by $\mathcal{A} \subset \{1, 2, \cdots, d\}$.*

The key tool we will use for this problem is two-sample hypothesis statistical tests and, in particular, a variant we define called the *conditional distribution hypothesis test*. We propose both model-free (*i.e.*, non-parametric) approach and model-based approaches to create this test. Importantly, we propose the novel use of a score-based test statistic for enabling more scalable hypothesis testing.

Unlike joint distribution shift detection, which cannot localize which features caused the shift, we define a new hypothesis test for each feature individually. Naïvely, the simplest test would be to check if the marginal distributions have changed for each feature (as explored by [25]); however, the marginal distribution would be easy for an adversary to simulate (*e.g.*, by looping the sensor values from a previous day). Thus, marginal tests are not sufficient for our purpose. Therefore, we propose to use conditional distribution tests. More formally, our null and alternative hypothesis for the $j$-th feature is that its full conditional distribution (*i.e.*, its distribution given all other features) has not shifted for all values of the other features.

**Definition 2. [Conditional Distribution Hypothesis Test]** *The feature shift hypothesis test is defined by the following null and alternative hypotheses:*

$$H_0 : \forall \boldsymbol{x}_{-j} \in \mathcal{X}_{-j}, \quad q(x_j|\boldsymbol{x}_{-j}) = p(x_j|\boldsymbol{x}_{-j}); \quad H_A : \exists \boldsymbol{x}_{-j} \in \mathcal{X}_{-j}, \quad q(x_j|\boldsymbol{x}_{-j}) \neq p(x_j|\boldsymbol{x}_{-j}).$$

Essentially, this test checks for any discrepancy in prediction of one feature given all the other features. Thus, if there are probabilistic dependencies between sensors (*e.g.*, two vibration sensors near each other), then any independent manipulation of a subset of sensors will be noticeable from a joint distribution perspective. We note that a special case of this test is the marginal distribution test for each feature. We now define a simple class of statistics that could be used to perform these tests.

**Definition 3. [Expected Conditional Distance]** *The* expected conditional distance (ECD) *test statistic is defined as follows:* $\gamma = \mathbb{E}_{\boldsymbol{x}_{-j}}[\phi(p(x_j|\boldsymbol{x}_{-j}), q(x_j|\boldsymbol{x}_{-j}))]$ *where* $\phi$ *is a statistical divergence between two distributions (*e.g.*, KL divergence, KS statistic, Anderson-Darling statistic).*

To estimate ECD from samples, there are two design questions: (1) How should the conditional distributions $p(x_j|\boldsymbol{x}_{-j})$ and $q(x_j|\boldsymbol{x}_{-j})$ be estimated? and (2) Which statistical divergence $\phi$ should be used? First, for question (1), we develop *model-free* (via KNN) and *model-based* (via density model) approaches for estimating the conditional distributions. An illustration of the similarities and differences between model-free and model-based can be seen in Fig. 2. Second, for question (2), we compare the common non-parametric Kolmogorov-Smirnov (KS) divergence statistic with a Fisher divergence that only requires computation of the score function (*i.e.*, the gradient of the log density with respect to the input).

**Model-free approach via k-nearest neighbors.** To avoid any modeling assumptions, we could approximate the conditional distribution with samples. We will only consider the estimation of the distribution distance $\phi$ given an $\boldsymbol{x}_{-j}$: $\phi(p(x_j|\boldsymbol{x}_{-j}), q(x_j|\boldsymbol{x}_{-j})) \approx \phi(A_j(\boldsymbol{x}_{-j}; k), B_j(\boldsymbol{x}_{-j}; k))$ where $A_j(\boldsymbol{x}_{-j}; k) = \{z_j : \boldsymbol{z} \sim X, \boldsymbol{z}_{-j} \in \mathcal{N}_k(\boldsymbol{x}_{-j}; X)\}$, $B_j(\boldsymbol{x}_{-j}; k) = \{z_j : \boldsymbol{z} \sim Y, \boldsymbol{z}_{-j} \in \mathcal{N}_k(\boldsymbol{x}_{-j}; Y)\}$, $X$ and $Y$ denote the empirical distributions of $p$ and $q$ respectively, and $\mathcal{N}_k(\boldsymbol{x}_{-j}; X)$ denotes the set of $k$ nearest samples in the dataset $X$ when only considering $\boldsymbol{x}_{-j}$ (*i.e.*, ignoring $x_j$). If $\boldsymbol{x}$ is in a high-dimensional space, then k-nearest neighbors may be quite far from each other and thus may not approximate the conditional distribution well (see Fig. 2 for a 2D example). In our experiments, we found that the model-free approach to estimating conditional distributions is quite computationally expensive (see times in Table 1) and performs relatively poorly in comparison to the model-based approaches.

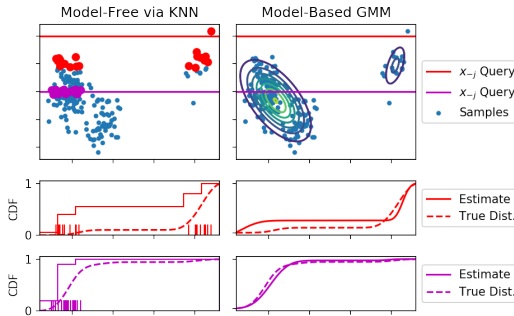

Figure 2: Conditional distributions (bottom) can be estimated using a model-free approach via KNN (left) or a model-based approach (right). Model-free could be better if true distribution is complex but may suffer from sparsity of data (especially in high dimensions) (notice red conditional distribution query point) whereas model-based will perform well if the data is sparse but may suffer if the modeling assumptions are violated.

**Model-based approach via explicit density models.** As an alternative to the computationally expensive model-free approach via KNN, we can consider a model-based approach that explicitly models the joint density function $\hat{p}(\boldsymbol{x})$ and computes the conditional distributions $\hat{p}(x_j \,|\, \boldsymbol{x}_{-j})$ based on the joint model. Ideally, we would be able compute the full conditionals, *i.e.*, $\hat{p}(x_j \,|\, \boldsymbol{x}_{-j})$ and be able to compute the $\phi$ efficiently (or even in closed-form). For Gaussian distributions, these conditional distributions can be computed in closed-form. For even more modeling power, we could estimate a normalizing flow such as MAF [23] or deep density destructors [13]. However, computing the conditional distributions $p(x_j|\boldsymbol{x}_{-j})$ of normalizing flows or recent deep density models is computationally challenging because, while they can explicitly compute the joint density function $p(\boldsymbol{x})$, they cannot provide us conditional densities easily. The simplest way to compute conditional densities from a deep density model is to compute a grid of joint density values along $x_j$ while keeping $\boldsymbol{x}_{-j}$ fixed and then renormalizing. To compute this for every dimension, this grid method would require $O(Md)$ evaluations of the deep density model where $M$ is the number of grid points

and $d$ is the number of dimensions. Additionally, even in the case where the conditional distribution can be computed in closed-form (*e.g.*, multivariate Gaussian or Gaussian-copulas), we still have to compute the relevant test statistic for every feature—thus roughly scaling as $O(d)$ tests. In the next section, we show how to circumvent this key difficulty by leveraging a score-based test statistic.

**Fisher divergence test statistic via score function.** We now consider the second question about which statistical divergence to use given an estimate of the conditional distributions. For the model-free approach, we only have samples so we must use a model-free test statistic such as the KS or Anderson-Darling (AD) test statistic. For the model-based approach, we propose to use another divergence $\phi$ that only requires the score function, which is defined as the gradient of the log density [11]:[2] $\psi(\boldsymbol{x}; p) \triangleq \nabla_{\boldsymbol{x}} \log p(\boldsymbol{x})$ . Traditionally, this score function has been used primarily for estimating distributions with intractable normalization constants—a method called *score matching*—because the score function is independent of the normalization constant [11]. The Fisher divergence can now be defined as the expected difference of the score function [21, 26, 28]:

$$\phi_{\text{Fisher}}(p, q) \triangleq \mathbb{E}_{p(\boldsymbol{x}) + q(\boldsymbol{x})}[\|\psi(\boldsymbol{x}; p) - \psi(\boldsymbol{x}; q)\|^2] = \mathbb{E}_{p(\boldsymbol{x}) + q(\boldsymbol{x})} \left[ \left\| \nabla_{\boldsymbol{x}} \log \frac{p(\boldsymbol{x})}{q(\boldsymbol{x})} \right\|^2 \right] \quad (1)$$

This divergence gives a way to measure the dissimilarity of distributions similar to the KL divergence: $\text{KL}(p, q) = \mathbb{E}_{p(\boldsymbol{x})}[\log \frac{p(\boldsymbol{x})}{q(\boldsymbol{x})}]$. If $\hat{p}$ and $\hat{q}$ are estimated models for $p$ and $q$ respectively, we can define the simple plug-in estimator: $\phi_{\text{Fisher}}(p, q) \approx \phi_{\text{Fisher}}(\hat{p}, \hat{q}) = \mathbb{E}_{\hat{p}(\boldsymbol{x}) + \hat{q}(\boldsymbol{x})}[\|\psi(\boldsymbol{x}; \hat{p}) - \psi(\boldsymbol{x}; \hat{q})\|^2]$ . We argue that this theoretically-motivated score-based statistic is desirable for the following reasons:

1. *Multiple feature test statistics simultaneously* - The joint score function can be used to trivially estimate the conditional score functions for all features at the same time because the conditional score function is merely the partial derivative (*i.e.*, a component of the gradient) of the log density function: $\psi(x_j; p_{x_j | x_{-j}}) = \frac{\partial}{\partial x_j}[\log p(x_j | \boldsymbol{x}_{-j})] = \frac{\partial}{\partial x_j} \left[ \log \frac{p(\boldsymbol{x})}{p(\boldsymbol{x}_{-j})} \right] = \frac{\partial}{\partial x_j} \log p(\boldsymbol{x}) = [\psi(\boldsymbol{x}; p)]_j$ , where the term with $p(\boldsymbol{x}_{-j})$ is constant w.r.t. $x_j$ and thus has a zero partial derivative—*i.e.*, the normalizing constant for conditional distributions can be ignored similar to the motivation for using scores for estimating unnormalized density models [11].
2. *Easy to compute for deep neural density and online learning models* - The score function is easy to compute for all features simultaneously using auto differentiation, even for complex deep density models (*e.g.*, [5, 13, 14, 24])—*i.e.*, only a single forward and backward pass is needed to compute all $d$ conditional score functions corresponding to each feature. This makes this approach very attractive for online learning where the deep density can be updated and a test statistic can be computed at the same time.

**Extension to time series.** We can extend this method to online time-series data in several straightforward ways. We can merely perform these hypothesis tests at regular intervals to determine when an attack has occurred as we will initially explore in our experiments; this enables the identification of when an attack occurs. Again, this will require fast and efficient methods for computing test statistics in an efficient manner so we choose to use the score-based Fisher divergence. Second, in Section 3.2 we introduce a time-dependent variant of bootstrapping to find thresholds which account for natural shifts across time. Lastly, it is fairly straightforward to extend our framework to account for some dependencies across time by modeling the difference between consecutive time points, *i.e.*, model $\delta_{\boldsymbol{x}} = \boldsymbol{x}^{(t)} - \boldsymbol{x}^{(t-1)}$, rather than the raw data points themselves. This could be further extended to second-order or higher-order differences and all these models could be cascaded for detecting shifts in the conditional distributions.

## 3  Experiments

**Attack Model.** Because we assume that the adversary can only observe and control a subset of sensors, an adversary cannot model or use the values of the other sensors to hide their attacks. Thus, if an attacker wants to avoid detection, the best they can do is to simulate the marginal distribution of the subset of sensors they control independent of the other sensors. Thus, a strong adversarial attack distribution (to avoid detection) given our assumption is a marginal distribution

attack defined as: MarginalAttack$(p, j) \triangleq p(\boldsymbol{x}_{-j})p(\boldsymbol{x}_j)$—notice that this forces $\boldsymbol{x}_j$ and $\boldsymbol{x}_{-j}$ to be independent under this attack distribution. Similarly, for a subset of sensors, we can define MarginalAttack$(p, \mathcal{A}) \triangleq p(\boldsymbol{x}_{\overline{\mathcal{A}}})p(\boldsymbol{x}_{\mathcal{A}})$. Practically, we implement this marginal attack by permuting the $j$-th column (or subset of columns) of $Y$ (samples from the $q$ distribution), which corresponds to sampling from the marginal distribution and breaks any dependencies between the other features. Also, we assume that once a sensor has been attacked, it will be attacked for all time points afterwards.

**Attack Strength and Difficulty.** We will define the *strength* of an attack in sensor networks as the number of observable and controllable sensors in the network. Compromising one sensor is the weakest attack model whereas compromising all sensors is the strongest attack model. Given our key observation that dependencies between features are critical for detecting sensor network attacks, we define the *difficulty* of the attack problem as the negative mutual information between compromised and uncompromised random variables of the true distribution. Formally, we define the attack strength and attack difficulty as: AttackStrength$(\mathcal{A}) \triangleq |\mathcal{A}|$, AttackDifficulty$(p_{\boldsymbol{x}}) \triangleq -\text{MI}(\boldsymbol{x}_{\mathcal{A}}, \boldsymbol{x}_{\overline{\mathcal{A}}})$, where $\mathcal{A} \subset \{1, 2, \cdots, d\}$ is the subset of sensors that are compromised. Intuitively, attack difficulty quantifies how well the uncompromised sensors should be able to predict the compromised sensors. On the one extreme, if all the sensors are completely independent of one another (*i.e.*, the mutual information is zero), then there is little hope of detecting an adversary who can mimic the marginal distributions via looping. On the other extreme, if all the sensors are equal to each other (*i.e.*, the mutual information is maximum), then even one observation could be used to determine which sensors are compromised (*i.e.*, any sensor that does not match the majority of other sensors).

**Method Details.** We compare three of the methods for estimating the expected conditional distance in Def. 3. First, we consider the model-free method based on KNN (§ 2) for estimating the conditional distributions paired with the non-parametric Kolmogorov-Smirnov (KS) statistic (denoted by "KNN-KS" in our results). To compare the model-free and model-based methods while keeping the statistic the same, we also consider a model-based method using a multivariate Gaussian to estimate both $p$ and $q$ and then computing the KS statistic by sampling $1,000$ samples from each model. Finally, we consider a model-based method using a multivariate Gaussian using the Fisher-divergence test statistic based on the score function (denoted "Score" or "MB-SM"). For the expectation over $\boldsymbol{x}_{-j}$ in Def. 3, we use 30 samples from both $X_{-j}$ and $Y_{-j}$ to empirically approximate this expectation. For all methods, we use bootstrap sampling to approximate the sampling distribution of the test statistic $\gamma$ for each of the methods above. In particular, we bootstrap $B$ two-sample datasets $\{(X_{\text{boot}}^{(b)}, Y_{\text{boot}}^{(b)})\}_{b=1}^{B}$ from the concatenation of the input samples $X$ and $Y$, *i.e.*, simulating the null hypothesis. For each $b$, we fit our model on $\{(X_{\text{boot}}^{(b)}, Y_{\text{boot}}^{(b)})\}^{(b)}$ (for Gaussians the fitting is fairly simple, but for deep density models we could update our model (1-2 epochs) with each $b$ rather than retraining), and estimate the expected conditional distance $\hat{\gamma}$ for all features (*i.e.*, $\hat{\gamma} = [\hat{\gamma}_1, \hat{\gamma}_2, \cdots, \hat{\gamma}_d]$). We set the target significance level to $\alpha = 0.05$ as in [25] (note: this is for the detection stage only; the significance level for the localization stage is not explicitly set). For multiple sensors, we use micro precision and recall which are computed by summing the feature-wise confusion matrices and then computing precision and recall, e.g., $\text{Prec}_{\text{micro}} = \frac{\sum_{i=1}^{d} \text{TP}_i}{\sum_{i=1}^{d} \text{TP}_i + \sum_{i=1}^{d} \text{FP}_i}$.

## 3.1 Simulated Experiments

**Simulation via Gaussian copula graphical models.** We generate data according to a Gaussian copula graphical model, denoted by $\mathcal{C}_{\mathcal{N}}(\boldsymbol{\mu}, \Sigma)$ where the non-zeros of $\Sigma^{-1}$ are the edges in the graphical model, paired with Beta marginal distributions with shape parameters $a = b = 0.5$. Note that the conditional dependency structure, i.e., the graphical model, and mutual information are the same as the corresponding multivariate Gaussian. However, the copula with Beta marginals distribution is very non-Gaussian since the Beta distribution is multimodal because the shape parameters $a$ and $b$ are both less than one. Thus, our model-based methods are not given significant advantage since $\mathcal{C}_{\mathcal{N}}$ has bimodal marginals with high densities on the edges of the distribution; which does not match the our assumed Gaussian distribution. We simulate relationships between 25 sensors in a real-world network using four graphical models: a complete graph, a chain graph, a grid graph, and a random graph using the Erdos-Renyi graph generation model with the probability of an edge set to 0.1 (illustration of the graph types in the Appendix). We set all the edge weights to be equal to a constant. We adjust this constant so that the attack difficulty for the middle sensor (*e.g.*, the mutual information between sensor 12 and all other sensors) is fixed at a particular value for each of the graphs to make them comparable.

Note that this is possible to compute in closed-form for multivariate Gaussian distributions—and Gaussian-copula distributions have equivalent mutual information because mutual information is invariant under invertible transformations. We select mutual information of $\{0.2, 0.1, 0.05, 0.01\}$ based on preliminary experiments that explore the limits of each method (*i.e.*, where they work reasonably well and where they begin to breakdown).

**Fixed single sensor.** We first investigate the simplest case where a user wants to know if a specific sensor in a network is compromised. For simulation, the first set of samples, *i.e.*, $X$, is assumed to be samples from the clean reference distribution $p$. However, $Y$ has two possible states: if $q$ is not compromised, then $Y$ also comes from the clean reference distribution, *i.e.*, $q = p$, but if $q$ is compromised, we assume that the samples come from a marginal distribution attack on sensor $j$, i.e., $q = \text{MarginalAttack}(p, j)$. We ran this experiment for all four graphs and different attack difficulties with 600 replications (300 with the $j$-th sensor compromised, 300 without compromise) using three random seeds (100 replications per case per seed). In the appendix, we give more results, and we can see that MB-SM outperforms the other two methods in terms of precision and recall and is also computationally most efficient. The best recall of 0.211 for 0.01 mutual information shows that we are testing the limits of our methods and detecting this attack is quite difficult.

**Unknown single sensor.** In this experiment, the user does not know *a priori* which single sensor is compromised, and thus the compromised sensor (if any) needs to be identified. We use the same simulation setup as in the fixed single sensor experiment. However, our detection method is split into two stages we call the *detection stage* and *localization stage* respectively. First, in the *detection stage*, we attempt to determine if any sensor is compromised. Second, for the *localization stage*, if the first stage detects

Table 1: Top: Precison and recall vs. mutual information with each divergence method for detecting the compromised single sensor, averaged over all four graphs. Marginal-KS refers to the state-of-the-art [25]. Bottom: wall-clock run time (seconds) per test.

| | Unknown Single Sensor | | | | | | | |
| | MB-SM | | MB-KS | | KNN-KS | | Marginal | |
| MI | Pre | Rec | Pre | Rec | Pre | Rec | Pre | Rec |
|---|---|---|---|---|---|---|---|---|
| 0.2 | **0.93** | **0.96** | 0.91 | 0.87 | 0.653 | 0.822 | 0.00 | 0.00 |
| 0.1 | **0.92** | **0.86** | **0.92** | 0.73 | 0.609 | 0.699 | 0.50 | 0.00 |
| 0.05 | **0.87** | **0.55** | 0.86 | 0.46 | 0.561 | 0.524 | 0.25 | 0.00 |
| 0.01 | **0.61** | **0.13** | 0.58 | 0.1 | 0.478 | 0.363 | 0.25 | 0.00 |
| Time | **0.044** | | 1.408 | | 2.246 | | 0.200 | |

an attack, we then attempt to localize which sensor was attacked. For the first stage, we report network has been compromised if *any* feature-level test is significant at the level $\alpha_{\text{bon}} = \frac{\alpha}{d}$, where $d$ is the number of sensors and $\alpha_{\text{bon}}$ is the conservative Bonferroni correction for multiple tests [3] which does not make any assumptions about the dependencies between tests. For the second stage (*i.e.*, if an attack is detected by stage one), we choose the sensor with the highest estimated expected conditional distance, *i.e.*, $\hat{j} = \arg\max_j \hat{\gamma}_j$. The positive class for computing precision and recall is defined as detecting an attack *and* identifying the correct sensor (*e.g.*, the method fails if it detects an attack but does not localize the correct sensor). In Table 1, we see that the score-based method had the best precision and recall for levels of mutual information. The as the other methods have lower recall, this suggests that these methods would miss many attacks. While the model-based KS method ("MB-KS") precision and recall are similar to score-based test statistic ("MB-SM"), the "MB-SM" computation time is over 30 times faster. Given the significant computational expense of the model-free KNN method and its overall poor performance compared to the model-based methods, we do not include the model-free KNN method in the other experiments.

**Unknown multiple sensors (Attack strength).** We consider the more complex case where multiple sensors could be compromised, *i.e.*, exploring the attack strength defined above. We assume we have some fixed budget of sensors that can be checked or verified, denoted by $k$, and we are seeking to identify which $k$ sensors have been compromised. We use the same setup as in the single sensor attack but attack three sensors via the marginal attack method. Similar to the unknown single sensor case, we proceed in a two stage fashion. The first stage is the same as the single sensor case where we use the Bonferroni correction to detect whether an attack has occurred. For the second stage, if an attack is predicted, the set of compromised sensors is predicted to be the top $k$ estimated statistics, i.e., $\widehat{\mathcal{A}} = \arg\max_{\{\mathcal{A}:\ |\mathcal{A}|=k\}} \sum_{j \in \mathcal{A}} \hat{\gamma}_j$—this is a simple generalization of the single sensor case. In the appendix, we show the precision and recall for the first detection stage for $k = 3$. It can be seen that with three compromised sensors, both the MB-SM and MB-KS perform well in stage one precision until a mutual information of 0.01, with the MB-SM again performing better in recall. In Table 2, we

show the micro precision and recall for the second localization stage for $k = 3$. As expected, the localization task (stage two) is significantly more difficult than merely identifying if an attack has occurred, with precision and recall usually less than 0.75. Yet again, we see that the score-based test statistic outperforms MB-KS in general—suggesting that the score-based test statistic is generally better for this task.

Table 2: Precision and Recall of the compromised sensor localization results using MB-SM or MB-KS with three attacked sensors.

| Graph: | Complete | | | | Cycle | | | | Grid | | | | Random | | | |
|---|---|---|---|---|---|---|---|---|---|---|---|---|---|---|---|---|
| Method: | SM | | MB-KS | | SM | | MB-KS | | SM | | MB-KS | | SM | | MB-KS | |
| MI \Metric: | Pre | Rec | Pre | Rec | Pre | Rec | Pre | Rec | Pre | Rec | Pre | Rec | Pre | Rec | Pre | Rec |
| 0.2 | **0.71** | **0.78** | 0.62 | 0.69 | **0.74** | **0.81** | 0.61 | 0.68 | **0.74** | **0.81** | 0.63 | 0.71 | **0.44** | **0.50** | 0.35 | 0.37 |
| 0.1 | **0.64** | **0.72** | 0.60 | 0.66 | **0.69** | **0.77** | 0.60 | 0.66 | **0.69** | **0.76** | 0.60 | 0.67 | **0.49** | **0.55** | 0.42 | 0.44 |
| 0.05 | **0.60** | **0.64** | 0.56 | 0.59 | **0.66** | **0.71** | 0.57 | 0.60 | **0.64** | **0.65** | 0.57 | 0.56 | **0.50** | **0.54** | 0.44 | 0.45 |
| 0.01 | **0.49** | **0.49** | 0.47 | 0.46 | **0.53** | **0.54** | 0.48 | 0.46 | **0.53** | **0.50** | 0.45 | 0.42 | **0.45** | **0.44** | 0.39 | 0.35 |

**Detecting and localizing time-series attack.** A strong use case for feature shift detection is detecting shifts in a time series. This involves not only finding which features have shifted, but also identifying *when* the shift happened. For this experiment, we sample according to our Gaussian copula model to produce a time series, which acts as a time series where the samples are temporally independent.

Table 3: Precision and Recall of localizing which sensors are compromised using the score-based test statistic with different window sizes and three attacked sensors.

| Graph: | Complete | | Cycle | | Grid | | Random | |
|---|---|---|---|---|---|---|---|---|
| Window | Prec | Rec | Prec | Rec | Prec | Rec | Prec | Rec |
| 200 | 0.29 | 0.21 | 0.69 | 0.81 | 0.46 | 0.53 | **0.38** | 0.48 |
| 400 | 0.41 | 0.52 | **0.70** | 0.88 | 0.54 | 0.73 | 0.34 | 0.47 |
| 600 | **0.49** | 0.70 | 0.64 | **0.92** | **0.61** | 0.80 | 0.27 | 0.48 |
| 800 | 0.48 | 0.73 | 0.59 | 0.89 | 0.53 | 0.80 | 0.32 | **0.50** |
| 1000 | 0.46 | **0.76** | 0.57 | 0.90 | 0.48 | **0.81** | 0.25 | 0.46 |

As in the other experiments, we fix $X \sim p$ to be our clean reference distribution, but now $Y \sim \mathcal{W}_K$ where $\mathcal{W}$ is a sliding window with a window size of $K$. We perform a stage one detection test sequentially on the time series every 50 samples (*i.e.*, each step is 50 samples). We choose 50 samples for simulation convenience but there is an inherent tradeoff between fidelity and computational time that could be explored. If an attack is detected at a time point, then $\mathcal{A}$ is predicted to be the $k$ greatest test statistics, $\hat{\gamma}_j$. This continues until the datastream has been exhausted of samples. For each experiment we compute a time-delay for the first detection, which we define to be $t_\Delta = t_{\text{det}} - t_{\text{comp}}$ where $t_{\text{comp}}$ is the first time step where a compromised sample enters $\mathcal{W}_K$, and $t_{\text{det}}$ is the step when an attack is detected. The first stage detection results can be seen in Table 4, where the shorter window-sizes (*e.g.*, $K = 200, 400$) have the highest precision, recall, and lowest delay for graphs other than random. However, in Table 3, it can be seen that the larger window sizes have better precision and recall in the second stage detection: finding which sensors are in $\mathcal{A}$. This highlights the trade-off between accuracy of first stage detection and that of second stage localization.

Table 4: Precision, Recall, and Time-Delay of detecting an attack using MB+SM with different window sizes and $|\mathcal{A}| = 3$ (i.e. three sensors are compromised).

| Graph: | Complete | | | Cycle | | | Grid | | | Random | | |
|---|---|---|---|---|---|---|---|---|---|---|---|---|
| Window | Prec | Rec | Delay | Prec | Rec | Delay | Prec | Rec | Delay | Prec | Rec | Delay |
| 200 | **0.87** | 0.63 | **2.33** | 0.85 | **1.00** | 1.00 | **0.81** | **0.93** | **1.67** | **0.81** | **1.00** | **0.00** |
| 400 | 0.80 | **1.00** | 2.67 | 0.80 | **1.00** | **0.67** | 0.74 | 1.00 | 2.33 | 0.72 | **1.00** | **0.00** |
| 600 | 0.69 | **1.00** | 2.33 | 0.70 | **1.00** | 1.00 | 0.76 | 1.00 | 2.67 | 0.56 | **1.00** | **0.00** |
| 800 | 0.66 | **1.00** | 3.33 | 0.63 | **1.00** | 1.00 | 0.66 | 1.00 | 3.00 | 0.64 | **1.00** | **0.00** |
| 1000 | 0.60 | **1.00** | 2.67 | 0.63 | **1.00** | 2.00 | 0.60 | 1.00 | 2.67 | 0.55 | **1.00** | **0.00** |

## 3.2 Experiments on Real-World Data

While our simulation experiments demonstrate the feature-shift detection task, we now test this task on real-world data. We present results on the UCI Appliance Energy Prediction dataset [4], UCI Gas Sensors for Home Activity Monitoring [10], and the number of new deaths from COVID-19 for the 10 states with the highest total deaths as of September 2020, measured by the CDC [1] (more dataset and experiment details can be found in the Apendix). Given the complex time dependencies among the data, we introduce a time-dependent bootstrapping method, "Time-Boot". Time-Boot subsamples random contiguous chunks from clean held out data (in real-world situations, this could be data which has already passed shift testing) to be the bootstrap distribution, as this accounts for time dependencies between data samples. In Table 5, we show our unknown sensor experiments with the original bootstrap ("MB-SM") and Time-Boot with the three datasets as well as results where the

Table 5: Detection (left) and localization (right) results for the UCI Appliance Energy Prediction, UCI Gas Sensor Array, and CDC's COVID-19 datasets, (top, middle, and bottom, respectively).

| Time Axis | MB-SM | | MB-SM-Time-Boot | | Deep-SM-Time-Boot | | MB-SM | | MS-SM-Time-Boot | | Deep-SM-Time-Boot | |
|---|---|---|---|---|---|---|---|---|---|---|---|---|
| | Prec | Recall | Prec | Recall | Prec | Recall | Prec | Recall | Prec | Recall | Prec | Recall |
| | *Feature shift detection (1st stage)* | | | | | | *Feature shift localization (2nd stage)* | | | | | |
| **Energy** | | | | | | | | | | | | |
| Unshuffled | 50.00% | **100%** | 16.67% | 8.86% | **55.00%** | 62.62% | 1.92% | **100%** | 2.38% | 1.27% | **3.00%** | 3.80% |
| Shuffled | 74.57% | **97.74%** | 75.25% | 96.20% | **77.89%** | 93.67% | 2.87% | **97.74%** | **75.25%** | 96.20% | 64.21% | 77.22% |
| **Gas** | | | | | | | | | | | | |
| Unshuffled | **50.00%** | **100%** | 11.11% | 2.27% | 22.22% | 4.55% | 8.85% | **100%** | **11.11%** | 2.27% | **11.11%** | 2.27% |
| Shuffled | 97.30% | **100%** | 75.86% | **100%** | 97.78% | **100%** | 51.43% | 68.35% | 56.90% | **75.00%** | 68.90% | 70.45% |
| **COVID-19** | | | | | | | | | | | | |
| Unshuffled | **50.00%** | **100%** | 0.00% | 0.00% | 0.0% | 0.0% | **6.96%** | **13.92%** | 0.00% | 0.00% | 0.00% | 0.00% |
| Shuffled | 66.67% | **88.61%** | 75.00% | 82.76% | **76.00%** | 65.52% | **51.43%** | **68.35%** | 40.62% | 44.83% | 36.00% | 31.03% |

ordered samples are shuffled throughout the whole dataset ("Time Axis Shuffled"), which creates a semi-synthetic dataset that breaks time dependencies between points, while keeping the real-world feature dependencies intact. The need for the time dependent bootstrap can be seen in the results of "MB-SM" Unshuffled as the 100% recall and 50% precision shows that the the method always predicts a shift. This is expected because the time-dependencies mean the data distribution has a natural (*benign*) shift over time. This highlights the inherent difficulty of detecting *adversarial* shifts in the presence of natural shifts: it is difficult (if not impossible in certain cases) to determine if a shift is benign or adversarial without further assumptions. Because our solution, MB-SM Time-Boot, takes into account some of these time-dependent shifts, it causes our detection threshold to be much higher, and thus leads to few detections (very low recall) as these adversarial shifts are hidden among the natural shifts. One possible solution to this would be to estimate time-dependent density models (e.g., autoregressive time models); however, exploration of time-dependent density models is outside the scope of this paper.

In Table 5 it can be seen that using a deep density model can improve the performance of our Time-Boot method. For the deep density model, we fit a normalizing flow using iterative Gaussianization [13, 15] as it is fast and stable because it only requires traditional PCA and histogram algorithms rather than end-to-end backpropagation. While recall is slightly reduced, our deep density method ("Deep-SM") improves the precision of both detection and localization compared to the Gaussian-based method ("MB-SM") for the Energy and Gas datasets, even when the time axis is unshuffled. For the COVID-19 data, since the data has strong benign shifts, our time aware bootstrapping method seems to set the detection threshold too high to detect the adversarial ones, and thus leads to 0% detection and consequently localization in the unshuffled case. Clearly, other deep density models including more general normalizing flows or autoregressive models could be used but we leave extensive comparisons to future work.

## 4 Discussion and Conclusion

In this paper, we introduced the problem of localizing the cause of a distribution shift to a specific set of features, which corresponds to a set of compromised sensors. We formalized this problem as performing multiple conditional hypothesis tests of the distribution in question. We develop both model-free and model-based approaches, including one using test statistic based on Fisher divergence. Our hypothesis test using this score-based test statistic performs both shift detection and localization using the model's gradient with respect to the input (which is usually already available for machine learning models). Using a realistic attack model, we then performed extensive experiments with simulated data and show the state-of-the-art performance of the score-based test statistic even with low levels of mutual information between features. Moreover, our results on three real-world datasets illuminate the difficulties of detecting adversarial shift in the presence of natural shifts in a time-series dataset, even when using a deep density model. Further experiments are needed to see how how to properly model time-dependent behaviors so that anomalous changes can be differentiated from natural time-based changes.

We hope our work suggests multiple natural extensions such as using an autoregressive model as seen in [7] to take better account for time-series data, or using density models that perform well in small data regimes thus allowing for better performance with smaller window sizes leading to faster detection. Another natural question is: if we have detected and even localized a shift, what do we do with this information? For example, automatic mitigation, validation checks for the localized feature shifts, or inferring the number of compromised sensors to be checked (rather than assuming a fixed budget). More broadly, the question this poses is can we characterize shifts beyond localization to give more information to system owners?

## Broader Impact

**Positive implications of our technology.** Our solution allows the deployment of sensor networks, including large-scale ones, with trust placed on the values obtained from the network. This can happen despite the presence of sophisticated adversaries, such as, an adversary that can observe and faithfully reproduce the values of some sensors. Such trust now opens up the possibility of deploying sensor networks in critical operations, such as, monitoring dangerous terrain (e.g., for IEDs), monitoring large industrial plants (e.g., for noxious gases), and monitoring industrial control systems through digital PLC controllers (e.g., for vibrations of motors). The huge volume of sensor network research has had challenges in translating to practical impact and one of the well-identified concerns has been that the networks are easy to compromise, thus eroding trust in their operation. Our work can start to address this concern.

Due to the fact that the compromised sensors can be localized, our work sheds some light on the reason behind its detection, allowing mitigation actions. This gets at the more explainable nature of ML models. Due to the scalable nature of our solution (e.g., the score based test statistic computation is lightweight), it is applicable to large-scale sensor networks.

**Negative implications of our technology.** An adoption of our solution without careful understanding of the adversary capabilities may lead to a false sense of confidence. For example, a sophisticated adversary may be able to compromise multiple correlated sensors and change their values in a correlated manner so as to defeat our defense. Generally, reasoning about such correlated attacks needs a higher degree of sophistication and is called upon prior to the use of our technology.

## Acknowledgments and Disclosure of Funding

Funding in direct support of this work was provided by: Northrop Grumman Corporation through the Northrop Grumman Cybersecurity Research Consortium (NGCRC), Army Research Lab through Contract number W911NF-2020-221, and National Science Foundation through their SaTC program, grant number CNS-1718637. There are no competing interests associated with this work.

## Footnotes

[1]The code for our experiments and methods is at `https://github.com/SeanKski/feature-shift`.

[2]More strictly, this is the traditional score function defined for Fisher information with respect to a hypothetical location parameter $\mu$ [11], *i.e.*, $\psi(\boldsymbol{x}; p) = \nabla_\mu p_\mu(\boldsymbol{x})|_{\mu=0} = \nabla_\mu p(\boldsymbol{x} + \mu)|_{\mu=0} = \nabla_{\boldsymbol{x}} p(\boldsymbol{x})$.

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
