[Supplementary Material]

# A  Experiment Details

Figure 3: Graphical models used for generating simulated data. (a) complete: all nodes directly dependent, (b) cycle: only adjacent nodes are directly dependent, (c) grid: other than edge nodes, all nodes have direct dependency with four other nodes, and (d) random (Erdos-Renyi): dependencies between nodes randomly generated (note: the random graph changes with every seed).

## A.1  Fixed Single Sensor

This experiment simulates where a users wants to know if a specific sensor in a network is compromised. Each experiment was run with MB-SM, MB-KS, KNN-KS, and Marginal-KS methods, for each of the four graph types, and each with decreasing mutual information, $MI \in \{0.2, 0.1, 0.05, 0.01\}$. These MI values were chosen via the breakdown points from previous empirical results (from where the models start performing poorly (0.2) to where the models perform very poorly (0.01)). Each experiment was ran with 600 replications (300 with $j$-th sensor compromised, 300 without compromise) using three random seeds (0, 1, 2). In each experiment, the $j$-th sensor which is attacked or not attacked is chosen randomly per replication, and detection success is measured solely on the chosen sensor in each replication. The results can be seen in Table 6 where it can be seen that the MB-SM method has the greatest recall for each MI. The Marginal-KS method has the greatest precision, but with a near zero recall. Excluding the Marginal-KS method, the MB-SM and MB-KS methods have similar levels of precision for each MI, with the MB-SM being the method with the lowest

computational time of them all. In each experiment the computational time is approximated to be $\frac{t_\gamma}{d}$ where $t_\gamma$ is the time to calculate the threshold, $\gamma$, for all sensors, and $d$ is the number of sensors. This assumption is reasonable since in practice only $\gamma_j$ needs to be calculated, thus dividing $\gamma$'s computational time by $d$.

## A.2 Unknown Single Sensor

This experiment explores the task of detecting if a sensor is compromised and if so, localizing which sensor has been taken over. The setup of the experiment is the same as the fixed sensor case, but with a different detection method, and the KNN-KS and Marginal-KS are not included due to expensive computational costs combined with low performance. The detection method consists of first seeing if any $\gamma_j$ is significant at $\alpha_{bon} = \frac{\alpha}{d}$ (i.e. if $\exists \gamma_j > \gamma_{0_j}$ where $\gamma_{0_j}$ is the $1 - \alpha_{bon}$ percentile of the $j$-th feature's conditional distance found from the bootstrapping step and $\alpha_{bon}$ is the Bonferroni correction for multiple tests). If any of the feature-level tests satisfy this condition, then a 'global attack' is detected. The attack is localized to a specific sensor by finding the sensor with the highest estimated conditional distance (i.e. $\hat{j} = \arg \max_j \hat{\gamma}_j$). The results for the localization of the unknown sensor can be seen in Table 1.

## A.3 Unknown Multiple Sensors

Here we consider the more complex case where multiple sensors could be compromised. We assume we are seeking to identify which $k$ sensors have been compromised. The setup for this experiment is the same as the unknown single sensor case, except we explore $k = \{2, 3, 4, 5\}$, and like-wise if an attack is detected, localize the compromised sensors with $|\mathcal{A}| = k$ (i.e. the top $k$ sensors with the highest conditional distance are predicted to be compromised). When a set of sensors are attacked, their compromised outputs sample from the joint marginal rather than from each individual sensor's marginal distribution. This is a much smarter attack, and can be more difficult to detect. The results are measured in two ways, the first is the detection results (i.e. detecting if a 'global attack' has occurred), and the second is localization results (i.e. if a 'global attack' was detected, how many of the predicted compromised sensors in $\hat{\mathcal{A}}$ actually compromised). This is done by computing a confusion matrix for each feature, and then summing along the feature axis (i.e. micro precision and recall). The detection results for $k = \{2, 3, 4, 5\}$ can be seen in Table 7 and Table 8 where it can be seen that the MB-SM outperforms the MB-KS in precision, recall, and as seen in the previous results, computational complexity. It can also be seen that there are slight improvements in detection as the number of compromised sensors increases, which makes sense as this means an attack should be easier to detect. The localization results can be seen in Table 9 and Table 10. Again the MB-SM outperforms the MS-KS in every category, except for the random graphs in which the MB-KS has higher precision and recall. The results show that as $k$ grows, the ability to localize which sensors are in $\mathcal{A}$ diminishes for both MB-KS and MB-SM. This makes sense as more and more conditional information is being destroyed (e.g. in the case of $d = 2$ if one sensor is compromised, it would be impossible to detect which is acting anomalously without any prior knowledge.)

## A.4 Detecting and localizing time-series attack

This experiment is to simulate detecting and localizing feature shift in time-series data. To do so, we sample 10,000 samples according to our Gaussian copula model to produce a time series where the samples are temporally independent. We set $X \sim p$ as our reference distribution. $Y \sim \mathcal{W}_\mathcal{K}$ is our query distribution where $\mathcal{W}_\mathcal{K}$ is a window of size $K = |X| = |Y|$ (unrelated to $k$ compromised sensors). $\mathcal{W}_\mathcal{K}$ slides over the time-series data with a step size of 50 (i.e. each step updates 50 new samples into and the 50 oldest samples out of the window), which was chosen as it is large enough to speed up calculations of the whole time series, but not too large that detection is strongly affected. The experiment was run for all graphs with MI=0.5 (in time-series experiments this is a less direct metric of problem hardness), with $k = \{1, 2, 3\}$, and the attack happening at 80% of the time-series (samples 8,000+). It should also be noted that the random selection of $\mathcal{A}$ (which sensors are compromised) only happens once, and $\mathcal{A}$ stays constant for the entire time of the attack. Within each window, the detection process is the same for the unknown multiple sensors experiment, with an additional metric introduced: time-delay. This metric measures how many steps of $\mathcal{W}_\mathcal{K}$ are taken between the time when an compromised sample enters the window and a attack is detected. More formally, $t_\Delta = t_{\text{det}} - t_{\text{comp}}$ where $t_{\text{comp}}$ is the first time step when a compromised sample enters $\mathcal{W}_K$,

and $t_{\text{det}}$ is the step when an attack is detected. The detection results can be seen in Table 11, where it can be seen that larger window sizes typically have better precision and recall when compared to smaller window sizes, however they also have longer time-delays (excluding $K = 200$ which can have very large time-delays due to poor detection overall). On average, the $K = 500$ has the best intersection of precision, recall, and time-delay. The localization results can be seen in Table 12 where again a clear trend of larger window-sizes ($K \geq 500$) have on average greater precision and recall than smaller window-sizes ($K < 500$). Choice of window-size will depend on the preferences of the user, but as a starting point, we suggest setting setting $K = 500$ for a balance of detection speed and localization accuracy. One thing we wish to explore further is how changing the step size affects these metrics as well.

## A.5  Experiments on Real-World Data

While our extensive simulation experiments were meant to demonstrate the feature-shift detection task, we also performed more experiments on real-world data. Each experiment consisted of detecting whether a *non-benign* distribution shift has happened, and if so, localizing it to a specific feature. This was performed on three datasets, the UCI Appliances Energy Prediction dataset [4], the UCI Gas sensors for home activity monitoring Data Set [10], and the CDC's United States COVID-19 Cases and Deaths by State over Time [1] as of late Sepetember, 2020. For the COVID-19 data, we extracted the number of new deaths per day for the 10 states with the greatest total number of deaths ('MI', 'PA', 'IL', 'NY', 'MA', 'FL', 'TX', 'CA', 'NJ', 'NYC') (note: the dataset lists New York state, 'NY' and New York City, 'NYC' as separate entities), and used a 2nd order polynomial to interpolate between each day with a resolution of $\frac{1}{2}$ hour. This resulted in an upsampling to $\sim 10{,}000$ samples.

We used two models for the experiments. The first was simply fitting $X$ and $Y$ to multivariate Gaussians using $\hat{p} = \mathcal{N}(\text{mean}(X), \text{var}(X))$ and likewise for $\hat{q}$. The second used a deep density model which fit a normalizing flow using iterative Gaussianization with the number of layers set to 2 for each $X$ and $Y$. The score function was used with both models, but two variants of bootstrapping were used. The first was the simple form of bootstrapping as explained in section 3, we will denote this as 'Simple Boot' for the remainder of this section. The second was 'Time Boot' which randomly subsampled contiguous time-series from held out clean data and set $(X, Y)$ to be the first and second half, respectively, of the subsampled data. More formally, $\left\{ X_{boot}^{(b)}, Y_{boot}^{(b)} \right\}_{b=1}^{B} = \left\{ T_{clean}[t^{(b)} - n : t^{(b)}], T_{clean}[t^{(b)} : t^{(b)} + n] \right\}_{b=1}^{B}$ where $B$ = number of bootstrap runs, $n$ is the number of samples in $X$ and $Y$, $T$ is the held out clean data, and $t^{(b)} \sim U[n, N_{clean} - n]$ is the $b^{(th)}$ split index uniformly sampled across the held out data (from $n$ to $N_{clean} - n$ to avoid boundary conditions and assure $X^{(b)}$ and $Y^{(b)}$ are the same size). As a preprocessing step for both bootstrapping methods, a first order approximation of the gradient of the samples w.r.t. time (i.e. $X_{t_i} = X_{t_{i+1}} - X_{t_i}$) was taken on the concatenation of $X$ and $Y$ in order to lessen some of the time-dependencies. Next, a power-transform (using Sklearn's power-transform with the Yeo-Johnson and standardizing set to true) was performed on the concatenation in order to make the data more Gaussian-like. Finally, the respective model was fit on the transformed $X$ and $Y$.

For experiments with 'Simple Boot' the number of trials was determined by $\lfloor \frac{N - 2n}{s} \rfloor$ where $N$ is the number of samples in the dataset, $n$ is the number of samples in $X$ and $Y$, and $s$ is the step size for stepping through the dataset. For experiments with 'Time Boot', the number of trials was similarly determined after half of the data was removed for our global time-based bootstrapping. $X$ and $Y$ were set via a sliding window through the dataset with size $2n$ and with a step size $s$. For half of the trials the Marginal Attack was performed on a random feature on a local copy of $Y$, and for the other half of the trials no attack was performed. For the experiments with 'Simple Boot', $\hat{p}$ and $\hat{q}$ were re-fit with each new $X$, $Y$ (i.e. each trial), and the hypothesis testing was performed with the threshold, $\gamma$ specific to that trial. In these experiments, we set $B_{Simple} = 50$ and $\alpha = 0.05$. Thus, due to time constraints the 'Simple-Boot' experiments were only performed using the score function with the multivariate Gaussian ('MB-SM'). For experiments with 'Time-Boot', a global threshold, $\gamma_{global}$ was used for the hypothesis testing. The threshold was determined using the held out data with $B_{time} = 500$ and $\alpha = 0.05$. The results for these experiments can be seen in Table 5.

Table 6: Top: precision and recall vs. mutual information with each divergence method for fixed single sensor, averaged over all four graphs. Bottom: wall-clock run time (seconds) per test (* fixed single index time is approximate).

| | Fixed Single Sensor | | | | | | | |
|---|---|---|---|---|---|---|---|---|
| | MB-SM | | MB-KS | | KNN-KS | | Marginal | |
| MI | Pre | Rec | Pre | Rec | Pre | Rec | Pre | Rec |
| 0.2 | 0.96 | **1.00** | 0.97 | 0.92 | 0.983 | 0.664 | **1.00** | 0.01 |
| 0.1 | 0.96 | **0.99** | 0.96 | 0.97 | 0.96 | 0.399 | **1.00** | 0.02 |
| 0.05 | 0.94 | **0.88** | 0.96 | 0.78 | 0.892 | 0.138 | **1.00** | 0.02 |
| 0.01 | 0.76 | **0.20** | 0.81 | 0.15 | 0.581 | 0.021 | **1.00** | 0.02 |
| Time* | **0.0018** | | 0.0563 | | 0.0898 | | 0.008 | |

Table 7: Precision and Recall of the compromised sensor detection results using MB-SM with $k = \{2, 3, 4, 5\}$ compromised sensors (out of 25 sensors total).

| | MB-SM | | | | | | | |
|---|---|---|---|---|---|---|---|---|
| | Complete | | | | | | | |
| K | 2 | | 3 | | 4 | | 5 | |
| MI | Precision | Recall | Precision | Recall | Precision | Recall | Precision | Recall |
| 0.2 | 0.920 | 0.997 | 0.930 | 1.000 | 0.880 | 1.000 | 0.923 | 1.000 |
| 0.1 | 0.922 | 0.980 | 0.910 | 1.000 | 0.904 | 0.998 | 0.919 | 1.000 |
| 0.05 | 0.912 | 0.913 | 0.920 | 0.940 | 0.912 | 0.963 | 0.920 | 0.992 |
| 0.01 | 0.856 | 0.720 | 0.870 | 0.750 | 0.862 | 0.762 | 0.872 | 0.798 |
| | Cycle | | | | | | | |
| K | 2.000 | | 3.000 | | 4.000 | | 5.000 | |
| MI | Precision | Recall | Precision | Recall | Precision | Recall | Precision | Recall |
| 0.2 | 0.946 | 1.000 | 0.940 | 1.000 | 0.909 | 1.000 | 0.898 | 1.000 |
| 0.1 | 0.943 | 0.995 | 0.930 | 1.000 | 0.916 | 1.000 | 0.924 | 1.000 |
| 0.05 | 0.923 | 0.944 | 0.930 | 0.970 | 0.914 | 0.966 | 0.915 | 0.989 |
| 0.01 | 0.824 | 0.731 | 0.850 | 0.770 | 0.855 | 0.784 | 0.870 | 0.793 |
| | Grid | | | | | | | |
| K | 2.000 | | 3.000 | | 4.000 | | 5.000 | |
| MI | Precision | Recall | Precision | Recall | Precision | Recall | Precision | Recall |
| 0.2 | 0.906 | 0.997 | 0.940 | 1.000 | 0.926 | 1.000 | 0.904 | 1.000 |
| 0.1 | 0.916 | 0.962 | 0.930 | 0.980 | 0.927 | 0.997 | 0.914 | 0.992 |
| 0.05 | 0.920 | 0.879 | 0.930 | 0.900 | 0.916 | 0.941 | 0.917 | 0.958 |
| 0.01 | 0.836 | 0.685 | 0.870 | 0.710 | 0.862 | 0.753 | 0.867 | 0.773 |
| | Random | | | | | | | |
| K | 2.000 | | 3.000 | | 4.000 | | 5.000 | |
| MI | Precision | Recall | Precision | Recall | Precision | Recall | Precision | Recall |
| 0.2 | 0.917 | 0.993 | 0.910 | 1.000 | 0.931 | 0.993 | 0.901 | 1.000 |
| 0.1 | 0.912 | 0.982 | 0.930 | 0.990 | 0.919 | 0.993 | 0.913 | 0.997 |
| 0.05 | 0.919 | 0.937 | 0.920 | 0.960 | 0.909 | 0.973 | 0.910 | 0.973 |
| 0.01 | 0.861 | 0.763 | 0.870 | 0.780 | 0.865 | 0.795 | 0.885 | 0.801 |

Table 8: Precision and Recall of the compromised sensor detection results using MB-KS with $k = \{2, 3, 4, 5\}$ compromised sensors (out of 25 sensors total).

| | MB-KS | | | | | | | |
|---|---|---|---|---|---|---|---|---|
| | **Complete** | | | | | | | |
| **K** | **2** | | **3** | | **4** | | **5** | |
| **MI** | **Precision** | **Recall** | **Precision** | **Recall** | **Precision** | **Recall** | **Precision** | **Recall** |
| 0.2 | 0.934 | 0.997 | 0.930 | 1.000 | 0.929 | 1.000 | 0.955 | 1.000 |
| 0.1 | 0.930 | 0.973 | 0.910 | 1.000 | 0.925 | 1.000 | 0.938 | 1.000 |
| 0.05 | 0.914 | 0.866 | 0.920 | 0.940 | 0.921 | 0.960 | 0.931 | 0.972 |
| 0.01 | 0.827 | 0.678 | 0.870 | 0.750 | 0.882 | 0.763 | 0.881 | 0.779 |
| | **Cycle** | | | | | | | |
| **K** | **2.000** | | **3.000** | | **4.000** | | **5.000** | |
| **MI** | **Precision** | **Recall** | **Precision** | **Recall** | **Precision** | **Recall** | **Precision** | **Recall** |
| 0.2 | 0.912 | 1.000 | 0.930 | 1.000 | 0.935 | 1.000 | 0.929 | 1.000 |
| 0.1 | 0.939 | 0.985 | 0.930 | 1.000 | 0.939 | 0.998 | 0.930 | 1.000 |
| 0.05 | 0.937 | 0.906 | 0.930 | 0.930 | 0.931 | 0.969 | 0.929 | 0.977 |
| 0.01 | 0.873 | 0.722 | 0.850 | 0.730 | 0.867 | 0.772 | 0.885 | 0.783 |
| | **Grid** | | | | | | | |
| **K** | **2.000** | | **3.000** | | **4.000** | | **5.000** | |
| **MI** | **Precision** | **Recall** | **Precision** | **Recall** | **Precision** | **Recall** | **Precision** | **Recall** |
| 0.2 | 0.927 | 0.977 | 0.910 | 1.000 | 0.925 | 0.990 | 0.920 | 1.000 |
| 0.1 | 0.930 | 0.905 | 0.920 | 0.980 | 0.924 | 0.977 | 0.921 | 0.995 |
| 0.05 | 0.909 | 0.779 | 0.910 | 0.850 | 0.901 | 0.883 | 0.928 | 0.900 |
| 0.01 | 0.840 | 0.604 | 0.840 | 0.660 | 0.833 | 0.698 | 0.866 | 0.721 |
| | **Random** | | | | | | | |
| **K** | **2.000** | | **3.000** | | **4.000** | | **5.000** | |
| **MI** | **Precision** | **Recall** | **Precision** | **Recall** | **Precision** | **Recall** | **Precision** | **Recall** |
| 0.2 | 0.927 | 0.847 | 0.900 | 0.920 | 0.369 | 0.343 | 0.927 | 0.970 |
| 0.1 | 0.928 | 0.875 | 0.920 | 0.930 | 0.450 | 0.435 | 0.925 | 0.980 |
| 0.05 | 0.925 | 0.826 | 0.910 | 0.900 | 0.486 | 0.440 | 0.933 | 0.947 |
| 0.01 | 0.868 | 0.662 | 0.870 | 0.720 | 0.431 | 0.346 | 0.883 | 0.768 |

Table 9: Precision and Recall of the compromised sensor localization results using MB-SM with $k = \{2, 3, 4, 5\}$ compromised sensors (out of 25 sensors total).

| | | | | | | | | |
|---|---|---|---|---|---|---|---|---|
| **MB-SM** | | | | | | | | |
| **Complete** | | | | | | | | |
| **K** | 2 | | 3 | | 4 | | 5 | |
| **MI** | **Precision** | **Recall** | **Precision** | **Recall** | **Precision** | **Recall** | **Precision** | **Recall** |
| 0.2 | 0.754 | 0.832 | 0.710 | 0.780 | 0.576 | 0.699 | 0.583 | 0.682 |
| 0.1 | 0.728 | 0.789 | 0.640 | 0.720 | 0.580 | 0.684 | 0.556 | 0.653 |
| 0.05 | 0.668 | 0.689 | 0.600 | 0.640 | 0.547 | 0.620 | 0.527 | 0.615 |
| 0.01 | 0.552 | 0.527 | 0.490 | 0.490 | 0.463 | 0.478 | 0.460 | 0.482 |
| **Cycle** | | | | | | | | |
| **K** | 2.000 | | 3.000 | | 4.000 | | 5.000 | |
| **MI** | **Precision** | **Recall** | **Precision** | **Recall** | **Precision** | **Recall** | **Precision** | **Recall** |
| 0.2 | 0.820 | 0.883 | 0.740 | 0.810 | 0.642 | 0.753 | 0.614 | 0.738 |
| 0.1 | 0.769 | 0.827 | 0.690 | 0.770 | 0.627 | 0.731 | 0.603 | 0.706 |
| 0.05 | 0.696 | 0.730 | 0.660 | 0.710 | 0.595 | 0.673 | 0.569 | 0.664 |
| 0.01 | 0.564 | 0.555 | 0.530 | 0.540 | 0.497 | 0.524 | 0.485 | 0.516 |
| **Grid** | | | | | | | | |
| **K** | 2.000 | | 3.000 | | 4.000 | | 5.000 | |
| **MI** | **Precision** | **Recall** | **Precision** | **Recall** | **Precision** | **Recall** | **Precision** | **Recall** |
| 0.2 | 0.788 | 0.883 | 0.740 | 0.810 | 0.662 | 0.763 | 0.611 | 0.730 |
| 0.1 | 0.764 | 0.818 | 0.690 | 0.760 | 0.646 | 0.741 | 0.598 | 0.701 |
| 0.05 | 0.704 | 0.698 | 0.640 | 0.650 | 0.596 | 0.658 | 0.564 | 0.639 |
| 0.01 | 0.554 | 0.528 | 0.530 | 0.500 | 0.498 | 0.508 | 0.480 | 0.498 |
| **Random** | | | | | | | | |
| **K** | 2.000 | | 3.000 | | 4.000 | | 5.000 | |
| **MI** | **Precision** | **Recall** | **Precision** | **Recall** | **Precision** | **Recall** | **Precision** | **Recall** |
| 0.2 | 0.492 | 0.543 | 0.440 | 0.500 | 0.420 | 0.478 | 0.401 | 0.481 |
| 0.1 | 0.538 | 0.590 | 0.490 | 0.550 | 0.460 | 0.531 | 0.440 | 0.518 |
| 0.05 | 0.563 | 0.582 | 0.500 | 0.540 | 0.465 | 0.531 | 0.448 | 0.517 |
| 0.01 | 0.488 | 0.460 | 0.450 | 0.440 | 0.420 | 0.425 | 0.402 | 0.413 |

Table 10: Precision and Recall of the compromised sensor localization results using MB-KS with $k = \{2, 3, 4, 5\}$ compromised sensors (out of 25 sensors total).

| | | | | | | | | |
|---|---|---|---|---|---|---|---|---|
| **MB-KS** | | | | | | | | |
| **Complete** | | | | | | | | |
| **K** | 2 | | 3 | | 4 | | 5 | |
| **MI** | **Precision** | **Recall** | **Precision** | **Recall** | **Precision** | **Recall** | **Precision** | **Recall** |
| 0.2 | 0.658 | 0.715 | 0.620 | 0.690 | 0.570 | 0.655 | 0.544 | 0.615 |
| 0.1 | 0.638 | 0.681 | 0.600 | 0.660 | 0.557 | 0.643 | 0.531 | 0.612 |
| 0.05 | 0.597 | 0.583 | 0.560 | 0.590 | 0.527 | 0.588 | 0.508 | 0.574 |
| 0.01 | 0.473 | 0.442 | 0.470 | 0.460 | 0.442 | 0.452 | 0.442 | 0.449 |
| **Cycle** | | | | | | | | |
| **K** | 2.000 | | 3.000 | | 4.000 | | 5.000 | |
| **MI** | **Precision** | **Recall** | **Precision** | **Recall** | **Precision** | **Recall** | **Precision** | **Recall** |
| 0.2 | 0.632 | 0.706 | 0.610 | 0.680 | 0.568 | 0.648 | 0.561 | 0.652 |
| 0.1 | 0.642 | 0.686 | 0.600 | 0.660 | 0.569 | 0.645 | 0.548 | 0.636 |
| 0.05 | 0.615 | 0.610 | 0.570 | 0.600 | 0.543 | 0.604 | 0.535 | 0.608 |
| 0.01 | 0.498 | 0.466 | 0.480 | 0.460 | 0.463 | 0.469 | 0.457 | 0.472 |
| **Grid** | | | | | | | | |
| **K** | 2.000 | | 3.000 | | 4.000 | | 5.000 | |
| **MI** | **Precision** | **Recall** | **Precision** | **Recall** | **Precision** | **Recall** | **Precision** | **Recall** |
| 0.2 | 0.680 | 0.730 | 0.630 | 0.710 | 0.584 | 0.667 | 0.555 | 0.652 |
| 0.1 | 0.658 | 0.654 | 0.600 | 0.670 | 0.578 | 0.652 | 0.540 | 0.630 |
| 0.05 | 0.611 | 0.543 | 0.570 | 0.560 | 0.535 | 0.565 | 0.521 | 0.551 |
| 0.01 | 0.491 | 0.411 | 0.450 | 0.420 | 0.441 | 0.433 | 0.432 | 0.425 |
| **Random** | | | | | | | | |
| **K** | 2.000 | | 3.000 | | 4.000 | | 5.000 | |
| **MI** | **Precision** | **Recall** | **Precision** | **Recall** | **Precision** | **Recall** | **Precision** | **Recall** |
| 0.2 | 0.369 | 0.343 | 0.350 | 0.370 | 0.356 | 0.385 | 0.366 | 0.413 |
| 0.1 | 0.450 | 0.435 | 0.420 | 0.440 | 0.403 | 0.440 | 0.403 | 0.461 |
| 0.05 | 0.486 | 0.440 | 0.440 | 0.450 | 0.424 | 0.446 | 0.419 | 0.458 |
| 0.01 | 0.431 | 0.346 | 0.390 | 0.350 | 0.385 | 0.354 | 0.384 | 0.367 |

Table 11: Precision and Recall of detecting which sensors are compromised using the score method with different window sizes (WS) and $k = \{1, 2, 3\}$.

**MB-SM**

**Complete**

| K | 1 | | | 2 | | | 3 | | |
|---|---|---|---|---|---|---|---|---|---|
| WS | Pre | Rec | Delay | Prec | Rec | Delay | Prec | Rec | Delay |
| 200 | 0.67 | 0.13 | 18 | 0.84 | 0.34 | 7 | 0.87 | 0.63 | 7 |
| 300 | 0.70 | 0.52 | 12 | 0.74 | 0.82 | 5 | 0.82 | 0.93 | 4 |
| 400 | 0.85 | 0.82 | 25 | 0.82 | 0.95 | 6 | 0.80 | 1.00 | 8 |
| 500 | 0.80 | 0.91 | 8 | 0.82 | 0.99 | 18 | 0.62 | 1.00 | 7 |
| 600 | 0.80 | 0.97 | 15 | 0.75 | 1.00 | 10 | 0.69 | 1.00 | 7 |
| 700 | 0.80 | 0.96 | 16 | 0.74 | 1.00 | 16 | 0.70 | 1.00 | 8 |
| 800 | 0.68 | 1.00 | 23 | 0.69 | 1.00 | 12 | 0.66 | 1.00 | 10 |
| 900 | 0.69 | 1.00 | 20 | 0.66 | 1.00 | 8 | 0.67 | 1.00 | 12 |
| 1000 | 0.66 | 1.00 | 20 | 0.60 | 1.00 | 9 | 0.60 | 1.00 | 8 |

**Cycle**

| K | 1 | | | 2 | | | 3 | | |
|---|---|---|---|---|---|---|---|---|---|
| WS | Pre | Rec | Delay | Prec | Rec | Delay | Prec | Rec | Delay |
| 200 | 0.95 | 0.99 | 7 | 0.89 | 0.91 | 6 | 0.85 | 1.00 | 3 |
| 300 | 0.75 | 1.00 | 6 | 0.82 | 1.00 | 4 | 0.79 | 1.00 | 4 |
| 400 | 0.80 | 1.00 | 5 | 0.80 | 1.00 | 6 | 0.80 | 1.00 | 2 |
| 500 | 0.66 | 1.00 | 4 | 0.63 | 1.00 | 4 | 0.68 | 1.00 | 3 |
| 600 | 0.74 | 1.00 | 5 | 0.68 | 1.00 | 2 | 0.70 | 1.00 | 3 |
| 700 | 0.69 | 1.00 | 3 | 0.66 | 1.00 | 6 | 0.64 | 1.00 | 5 |
| 800 | 0.56 | 1.00 | 3 | 0.67 | 1.00 | 6 | 0.66 | 1.00 | 3 |
| 900 | 0.63 | 1.00 | 4 | 0.57 | 1.00 | 8 | 0.61 | 1.00 | 5 |
| 1000 | 0.64 | 1.00 | 6 | 0.58 | 1.00 | 5 | 0.63 | 1.00 | 6 |

**Grid**

| K | 1 | | | 2 | | | 3 | | |
|---|---|---|---|---|---|---|---|---|---|
| WS | Pre | Rec | Delay | Prec | Rec | Delay | Prec | Rec | Delay |
| 200 | 0.74 | 0.26 | 21 | 0.84 | 0.52 | 9 | 0.81 | 0.93 | 5 |
| 300 | 0.77 | 0.72 | 12 | 0.80 | 0.82 | 7 | 0.90 | 1.00 | 8 |
| 400 | 0.77 | 0.81 | 22 | 0.78 | 0.97 | 12 | 0.74 | 1.00 | 7 |
| 500 | 0.79 | 1.00 | 12 | 0.81 | 1.00 | 6 | 0.69 | 1.00 | 7 |
| 600 | 0.79 | 1.00 | 16 | 0.64 | 1.00 | 8 | 0.75 | 1.00 | 8 |
| 700 | 0.72 | 1.00 | 14 | 0.66 | 1.00 | 12 | 0.64 | 1.00 | 4 |
| 800 | 0.74 | 1.00 | 18 | 0.76 | 1.00 | 11 | 0.66 | 1.00 | 9 |
| 900 | 0.75 | 1.00 | 26 | 0.65 | 1.00 | 12 | 0.63 | 1.00 | 6 |
| 1000 | 0.68 | 1.00 | 12 | 0.66 | 1.00 | 10 | 0.60 | 1.00 | 8 |

**Random**

| K | 1 | | | 2 | | | 4 | | |
|---|---|---|---|---|---|---|---|---|---|
| WS | Pre | Rec | Delay | Prec | Rec | Delay | Prec | Rec | Delay |
| 200 | 0.88 | 1.00 | 0 | 0.88 | 1.00 | 1 | 0.81 | 1.00 | 0 |
| 300 | 0.78 | 1.00 | 0 | 0.78 | 1.00 | 0 | 0.83 | 1.00 | 0 |
| 400 | 0.73 | 1.00 | 1 | 0.72 | 1.00 | 0 | 0.72 | 1.00 | 0 |
| 500 | 0.70 | 1.00 | 1 | 0.71 | 1.00 | 0 | 0.60 | 1.00 | 0 |
| 600 | 0.57 | 1.00 | 0 | 0.67 | 1.00 | 0 | 0.56 | 1.00 | 0 |
| 700 | 0.67 | 1.00 | 1 | 0.64 | 1.00 | 0 | 0.65 | 1.00 | 0 |
| 800 | 0.63 | 1.00 | 1 | 0.64 | 1.00 | 0 | 0.64 | 1.00 | 0 |
| 900 | 0.54 | 1.00 | 1 | 0.60 | 1.00 | 0 | 0.59 | 1.00 | 0 |
| 1000 | 0.61 | 1.00 | 1 | 0.55 | 1.00 | 1 | 0.55 | 1.00 | 0 |

Table 12: Precision and Recall of localizing which sensors are compromised using the score method with different window sizes (WS) and $k = \{1, 2, 3\}$.

| | | | | | | |
|---|---|---|---|---|---|---|
| **MB-SM** | | | | | | |
| **Complete** | | | | | | |
| **K** | 1 | | 2 | | 3 | |
| **WS** | **Precision** | **Recall** | **Precision** | **Recall** | **Precision** | **Recall** |
| 200 | 0.08 | 0.02 | 0.13 | 0.05 | 0.29 | 0.21 |
| 300 | 0.27 | 0.20 | 0.29 | 0.33 | 0.26 | 0.30 |
| 400 | 0.49 | 0.47 | 0.43 | 0.50 | 0.41 | 0.52 |
| 500 | 0.57 | 0.65 | 0.50 | 0.60 | 0.37 | 0.59 |
| 600 | 0.65 | 0.78 | 0.49 | 0.66 | 0.48 | 0.70 |
| 700 | 0.73 | 0.88 | 0.57 | 0.77 | 0.46 | 0.66 |
| 800 | 0.64 | 0.93 | 0.53 | 0.76 | 0.48 | 0.73 |
| 900 | 0.67 | 0.97 | 0.60 | 0.91 | 0.55 | 0.81 |
| 1000 | 0.66 | 0.99 | 0.54 | 0.89 | 0.46 | 0.76 |
| **Cycle** | | | | | | |
| **K** | 1 | | 2 | | 3 | |
| **WS** | **Precision** | **Recall** | **Precision** | **Recall** | **Precision** | **Recall** |
| 200 | 0.40 | 0.14 | 0.44 | 0.27 | 0.46 | 0.53 |
| 300 | 0.70 | 0.65 | 0.54 | 0.55 | 0.61 | 0.68 |
| 400 | 0.75 | 0.78 | 0.55 | 0.68 | 0.54 | 0.73 |
| 500 | 0.77 | 0.98 | 0.57 | 0.70 | 0.51 | 0.75 |
| 600 | 0.79 | 1.00 | 0.52 | 0.81 | 0.61 | 0.80 |
| 700 | 0.72 | 0.99 | 0.58 | 0.88 | 0.49 | 0.77 |
| 800 | 0.74 | 1.00 | 0.69 | 0.91 | 0.53 | 0.80 |
| 900 | 0.75 | 1.00 | 0.63 | 0.98 | 0.49 | 0.77 |
| 1000 | 0.68 | 1.00 | 0.61 | 0.93 | 0.48 | 0.81 |
| **Grid** | | | | | | |
| **K** | 1 | | 2 | | 3 | |
| **WS** | **Precision** | **Recall** | **Precision** | **Recall** | **Precision** | **Recall** |
| 200 | 0.95 | 0.99 | 0.74 | 0.76 | 0.69 | 0.81 |
| 300 | 0.75 | 1.00 | 0.68 | 0.83 | 0.68 | 0.86 |
| 400 | 0.80 | 1.00 | 0.69 | 0.86 | 0.70 | 0.88 |
| 500 | 0.66 | 1.00 | 0.54 | 0.86 | 0.61 | 0.90 |
| 600 | 0.74 | 1.00 | 0.61 | 0.89 | 0.64 | 0.92 |
| 700 | 0.69 | 1.00 | 0.58 | 0.88 | 0.58 | 0.91 |
| 800 | 0.56 | 1.00 | 0.60 | 0.88 | 0.59 | 0.89 |
| 900 | 0.63 | 1.00 | 0.51 | 0.89 | 0.54 | 0.89 |
| 1000 | 0.64 | 1.00 | 0.51 | 0.88 | 0.57 | 0.90 |
| **Random** | | | | | | |
| **K** | 1 | | 2 | | 4 | |
| **WS** | **Precision** | **Recall** | **Precision** | **Recall** | **Precision** | **Recall** |
| 200 | 0.85 | 0.98 | 0.57 | 0.65 | 0.38 | 0.47 |
| 300 | 0.78 | 1.00 | 0.52 | 0.67 | 0.44 | 0.53 |
| 400 | 0.73 | 1.00 | 0.48 | 0.67 | 0.34 | 0.47 |
| 500 | 0.70 | 1.00 | 0.47 | 0.67 | 0.31 | 0.51 |
| 600 | 0.57 | 1.00 | 0.45 | 0.67 | 0.27 | 0.48 |
| 700 | 0.67 | 1.00 | 0.43 | 0.67 | 0.32 | 0.49 |
| 800 | 0.63 | 1.00 | 0.43 | 0.67 | 0.32 | 0.50 |
| 900 | 0.54 | 1.00 | 0.40 | 0.67 | 0.28 | 0.48 |
| 1000 | 0.61 | 1.00 | 0.37 | 0.67 | 0.25 | 0.46 |