[Reviews · NeurIPS 2020]

Review 1

Summary and Contributions: The paper proposes a feature shift detection algorithm based on conditional distribution tests.

Strengths: The problem is interesting and novel.

Weaknesses: More experiments based on real applications is required to justify the effectiveness of the proposed method. - The experiments on real world data is insufficient and the result seems bad compare to simulation. More explanation is required. - In table 1, please explain why Marginal-KS is extremely bad? - In table 1, what the first column Rec represent? I've read the response and comments from other reviewers. The response well addressed my concern as the authors add more real world experiments and it seems promising. Thus, I will increase my score.

Correctness: Yes

Clarity: Yes

Relation to Prior Work: Yes

Reproducibility: Yes

Additional Feedback:


Review 2

Summary and Contributions: This manuscript proposes to detect feature shift during observation of a multivariate signal. In particular, the authors propose a novel approach based on score function. This approach is appealing for its computational aspects and the possibility of relying on flexible generative models for fitting the data. A set of experiments, focusing on a compelling time-series setting, shows that the model actually identifies which covariates shifted.

Strengths: This manuscript posits a novel and interesting extension of the outlier detection problem, with added interpretability constraints where one needs to identify which latent variables shifted. The proposed method is also simple and appealing, with the requirement to fit only one multivariate black-box density model (and not one per hypothesis). The simulated experiments based on multivariate Gaussian distributions are well led and are compelling.

Weaknesses: The manuscript highly markets the possibility of using this method for arbitrary dependence structure, in particular including those modelled with a deep density model (e.g., normalizing flows). It is mentioned in the abstract, in the introduction and in Section 2. However, this does not appear in the experiments (even the real-world data case appears to be treated with multivariate Gaussian). The discussion about statistical significance merits to be further extended. Line 255 - 258, it is mentioned that the bootstrap is used for simulating the null hypothesis. + Does that mean that the model must be fit multiple times, or that only the score function must be calculated with respect to these bootstrap datasets? If this is just the score function, why do we expect the density to be accurate in those areas, especially in the case of a deep density model? + How does this method perform at controlling the False Discovery Rate?

Correctness: What is in this manuscript seems reasonable to me

Clarity: The paper is well written and clear. Minor points: KNN is not defined before being used A space is missing in line 175 Words are missing or added in line 222 and 224 Typo line 248 Missing reference line 356

Relation to Prior Work: To my knowledge, related work is cited

Reproducibility: Yes

Additional Feedback: After author feedback ------ I would like to thank the author for running more experiments, in particular for the deep density model. I think these make the paper stronger and more relevant!


Review 3

Summary and Contributions: The paper studies the question of which features lead to the distribution shift. They formalize this problem multiple conditional distribution hypothesis tests and propose both non-parametric and parametric statistical tests. In particular, they build on the idea of density model score function to build flexible statistics.

Strengths: The paper studies an important problem of attributing distribution shift to specific features. The formulation of this important task into a statistical problem of multiple conditional distribution hypothesis tests opens the door to many existing algorithms in conditional testing. The resulting proposal hence leverage this connection and utilize a computationally efficient density model score function. Notably, this statistic and be computed for all dimensions in a single forward and backward pass. Moreover, it inherits the flexibility of current density estimators. The fomulation of the task of distribution shift attribution is an interesting and important contribution. The development of a computationally efficient test statistic makes it applicable to model applications in complicated settings.

Weaknesses: While the test statistic is computationally tractable and flexible, it is unclear how the use of flexible density estimators may affect the power of the tests. In particular, the proposed statistic is compatible with any density model including deep density models such as normalizing flows or autoregressive models. However these flexible density models are known for requiring a large number of samples to produce good density estimation. In such cases, it may decrease the power of the statistical tests for distribution shift attribution when the sample size is limited. This aspect of using flexible density estimators is worth discussing in the paper. The paper also has extended the proposal to a time-series setting. However, the results in tables 3 and 4 appear quite sensitive to the choice of window size. A discussion of how to choose window size for the proposal in time-series settings would be very helpful, especially such distribution shift task commonly occur in a time-series setting.

Correctness: The paper appears correct.

Clarity: The paper is quite well-written.

Relation to Prior Work: The paper adequately discussed prior work.

Reproducibility: Yes

Additional Feedback: See above. -------------- Thank you to the authors for the rebuttal. I have read the rebuttal and my evaluation stays the same.


Review 4

Summary and Contributions: The paper addresses distribution shift detection, casting it as a conditional shift problem, that is designed for multivariate settings. Through the use of the density model score function, an efficient algorithm is given that uses just a single forward and backward pass, and can be combined with modern density models based on neural networks. A key differentiator for this work is the desire to localise a shift (e.g. which sensor in a sensor network) as well as detecting the shift. ========= Post rebuttal: It's commendable that the authors ran more experiments using deep density models, in response to all reviewers. Pleasingly, the Deep-SM model seems to do even better, although I found the table in the rebuttal a little hard to parse. The authors also answered my technical points satisfactorily. I've raised my score accordingly

Strengths: - Neat application of the score function method to statistical testing via the Fisher divergence - Attack model is well constructed, and the range of Gaussian copula models used in the simulation study is well thought out

Weaknesses: - The KNN approach to building a conditional density seems slightly strange. It would seem that other non-parametric approaches, such as K-D trees, might be better suited to this task - On of the purported advantages of the score function approach is the ability to use modern density models. It’s therefore a pity that these aren’t used in the paper, for example neural-kernelized conditional density estimation [1] or methods in [2]. - Real world experiments are very preliminary [1] Sasaki, Hiroaki, and Aapo Hyvärinen. "Neural-kernelized conditional density estimation." arXiv preprint arXiv:1806.01754 (2018). [2] Rothfuss, Jonas, et al. "Conditional density estimation with neural networks: Best practices and benchmarks." arXiv preprint arXiv:1903.00954 (2019).

Correctness: Method is correct. Empirical methodology seems solid.

Clarity: Mostly well written and clear. There's no conclusions section, presumably due to lack of space, which together with the brief discussion of real-world experiments, gives an "unfinished" feel to the paper. In the model free approach, I can see that A and B are describing if the sample is in the nearest neighbours from both p and q, but what is the distance function phi? Is it simply the indicator function?

Relation to Prior Work: The related work on shift detection is well described. The novelties in terms of the score function-based inference and the localization of shifts are clearly positioned relative to previous works.

Reproducibility: Yes

Additional Feedback: KS - expand on first use (L185) Duplicate citations [19] & [20]

[Author Response · NeurIPS 2020]

We thank the reviewers for their very helpful comments and suggestions. We will first address the primary concerns about real-world experiments and deep density models and then answer detailed comments and questions.

**More experiments on real world data (R1).** We present more experiments on both the UCI energy dataset and a UCI gas sensor dataset. Given the complexities of data with strong time dependencies, we walk through a few new real data experiments using a time-dependent bootstrap and a shuffled time axis (see Table A). First, we provide results for our unknown sensor experiment (the submission only contained results for a fixed single sensor) via the original bootstrap (denoted "MB-SM, Unshuffled"). The 100% recall but very poor precision (i.e., it always predicts a shift) is expected because the real data exhibits strong time dependencies, and thus *natural* (or "benign") distribution shifts exist across time. This highlights the inherent difficulty of detecting *adversarial* shifts in the presence of natural shifts: it is difficult (if not impossible in certain cases) to determine if a shift is benign or adversarial without further assumptions. As a first attempt to overcome this issue, we modify our bootstrapping method to account for natural shifts; specifically, we sample chunks of time jointly together (rather than completely random time points) from a held-out clean dataset. This causes our detection threshold to be much higher, and thus our time-dependent bootstrap method ("Time-Boot") rarely detects (very low recall) the adversarial shifts because they are hidden among natural shifts. Fundamentally, this is because our current density models assume no time dependency. One possible solution is to estimate time-dependent density models (e.g., autoregressive time models); however, exploration of time-dependent density models is outside the scope of this paper which focuses on the localization aspect. Thus, as a final experiment, we shuffle the dataset along the time axis so that all time dependencies are broken; this creates a semi-synthetic dataset because it retains the feature dependencies but removes time dependencies. In this semi-synthetic setting (denoted "Shuffled"), our method performs much better and similar to the simulation results. We hope these results and discussion bring insight into the challenges of time series data and encourage further work in this area.

**Experiments with deep density models (R1, R2, R3, R4).** In this experiment, we demonstrate that using a deep density model can improve the performance of our method. We fit a normalizing flow using iterative Gaussianization[1] which is fast and stable because it only requires iteratively estimating a PCA projections and univariate histograms and thus can be carefully controlled to help avoid overfitting. While recall is slightly reduced, our deep density method ("Deep-SM") significantly improves the precision of both detection and localization compared to the Gaussian-based method ("MB-SM") even when the time axis is unshuffled. Clearly, other deep density models including more general normalizing flows or autoregressive models could be used but we leave extensive comparisons to future work.

Table A: Unknown single sensor experiment with UCI appliance energy dataset (left) and UCI gas sensor dataset (right).

| Time Axis | MB-SM | | MB-SM-Time-Boot | | Deep-SM-Time-Boot | | MB-SM | | MB-SM-Time-Boot | | Deep-SM-Time-Boot | |
|---|---|---|---|---|---|---|---|---|---|---|---|---|
| | Prec | Recall | Prec | Recall | Prec | Recall | Prec | Recall | Prec | Recall | Prec | Recall |
| *Feature shift detection (1st stage)* | | | | | | | | | | | | |
| Unshuffled | 50.00% | **100%** | 16.67% | 8.86% | **55.00%** | 62.62% | **50.00%** | **100%** | 11.11% | 2.27% | 22.22% | 4.55% |
| Shuffled | 74.57% | **97.74%** | 75.25% | 96.20% | **77.89%** | 93.67% | 97.30% | **100%** | 75.86% | **100%** | **97.78%** | **100%** |
| *Feature shift localization (2nd stage)* | | | | | | | | | | | | |
| Unshuffled | 1.92% | **100%** | 2.38% | 1.27% | **3.00%** | 3.80% | 8.85% | **100%** | **11.11%** | 2.27% | **11.11%** | 2.27% |
| Shuffled | 2.87% | **97.74%** | **75.25%** | 96.20% | 64.21% | 77.22% | 4.52% | **100%** | 56.90% | 75.00% | **68.90%** | 70.45% |

**R1.** *Table 1 clarifications.* There was a typo and the first column should match that of the Table 2 which represents the difficulty of the attack based on mutual information between variables, see L232 - L245 on "Attack Strength and Difficulty". Marginal-KS is very bad because the attack model is very strong, i.e., it mimics the marginal distribution of the sensor. Thus, marginal KS will naturally fail—highlighting the limitation of prior work for this adversarial attack.

**R2.** *For bootstrapping, does the model need to be fit multiple times?* Yes, we refit the model for every bootstrap iteration. For Gaussian, this is fairly simple. For deep density models, we could train one model on all the data first, and then update the model slightly (1-2 epochs) for each bootstrap (similar to transfer learning). *How does bootstrapping perform at controlling the False Discovery Rate?* For the detection stage, the FDR was controlled below 0.05 in all but the hardest cases, see Table 1 (note that FDR is 1 - Precision). See also Table 6 and 7 in appendix. For the more challenging localization stage, we do not explicitly control the FDR.

**R3.** *Deep density with limited samples.* Thanks for the comment. Our results above demonstrate that deep models can indeed be helpful though we will add some discussion regarding this challenge. *Choice of window size.* Thanks for the comment. We highlight that the choice of window size is a trade-off between the delay in detecting a shift (Table 4) and the error of the sensor localization (Table 3). Also, the particular application may have resource constraints.

**R4.** *Motivation for KNN approach.* We wanted a method that could compute a "conditional" KS statistic since the marginal KS statistic is well-known for detecting 1D shifts. We are unsure what is meant by "K-D trees"; could you point to a reference for this? *Neural-kernelized and other conditional density estimation.* Thanks for the pointers. We note that using conditional density estimation would require estimating a different conditional density models for every feature. In contrast, a *single* joint density model can be used to compute all conditional statistics via the score function.

## Footnotes

[1] [1] V. Laparra, G. Camps-Valls, and J. Malo. Iterative Gaussianization: From ICA to random rotations. *IEEE Tran. on Neural Networks*, 2011. [2] D. I. Inouye and P. Ravikumar. Deep density destructors. In *ICML*, 2018.


[Meta-Review · NeurIPS 2020]

Four knowledgeable referees support acceptance for the contributions, notably a method for detecting feature shift, and I also recommend acceptance. The problem formulation itself is a contribution, and the proposed method is well-motivated and performs well empirically.